# The Impact of Different Types of Exercise on Executive Functions in Overweight/Obese Individuals: A Systematic Review and Network Meta-Analysis

**DOI:** 10.3390/bs14121227

**Published:** 2024-12-19

**Authors:** Jia Guo, Jingqi Liu, Ruihan Zhu, Guochun Liu, Man Zheng, Chunmei Cao

**Affiliations:** 1Division of Sports Science and Physical Education, Tsinghua University, Beijing 100084, China; guojia21@mails.tsinghua.edu.cn (J.G.); zrh22@mails.tsinghua.edu.cn (R.Z.); lgc890206@163.com (G.L.); zhengm20@mails.tsinghua.edu.cn (M.Z.); 2State Key Laboratory of Cognitive Neuroscience and Learning, Beijing Normal University, Beijing 100091, China; jingqiliu@mail.bnu.edu.cn

**Keywords:** exercise, overweight, obese, network meta-analysis, executive function, inhibitory control

## Abstract

To compare the effects of different exercise training on executive function (EF) in obese or overweight individuals. PubMed, Web of Science, SPORTDiscus, MEDLINE, and CINAHL. The included articles, in English, should have been published from January 2000 to February 2024. All included studies were randomized controlled trials (RCTs) of exercise intervention in overweight or obese populations. The primary outcomes are EFs, which encompass core functions (e.g., inhibitory control, working memory (WM), and cognitive flexibility (CF)) and higher-level functions (e.g., responding, planning, and problem-solving). Therefore, the primary outcomes should include at least one of the above indicators. Additionally, given the focus of many exercise intervention studies on academic performance (AP) in obese adolescents, a secondary outcome includes AP. This meta-analysis synthesizes findings from 20 RCTs published between 2010 and 2023, encompassing a total of 1183 overweight or obese participants. Interventions were categorized into seven types: control training (CT), aerobic exercise (AE), resistance training (RT), coordinated physical activity (CPA), prolonged time of exercise (PTE), high-intensity interval training (HIIT), and AE combined with RT (mix mode, MIX). The surface under the cumulative ranking curve (SUCRA) results demonstrated the preferable effects of various interventions on EF improvement. SUCRA values indicate that CPA performs best in improving the accuracy and reaction speed of CF, as well as the reaction speed of inhibitory control in children and adolescents. AE shows significant effects in enhancing AP in this population. Additionally, PTE excels in improving CF and inhibitory control in middle-aged and older adults. Through subgroup analysis based on age and intervention duration, we found AE exhibited a significant effect on interventions for the 0–17 age group (SMD = 0.84, 95%CI = 0.31~1.38, *p* = 0.002) and interventions lasting 8–16 weeks showed significant improvement in EFs (SMD = 0.53, 95%CI = 0.00~1.05, *p* = 0.048). There was also a significant difference between CPA intervention and CT (SMD = 1.12, 95%CI = 0.45~1.80, *p* = 0.001) in children and adolescents. Additionally, PTE showed significant effects for middle-aged adults aged 17–59 (SMD = 0.93, 95%CI = 0.11~1.96, *p* < 0.027). Conclusions: This NMA found that CPA and AE have significant benefits for CF, inhibitory control, and AP in children and adolescents. Furthermore, PTE improves EFs in adults and older adults. Combining the findings of this study with previous related research, we recommend that OW/OB begin by interrupting prolonged sedentary behavior and increasing fragmented physical activity, gradually incorporating AE, RT, and CPA (such as jump rope).

## 1. Introduction

Currently, the incidence of overweight and obesity among populations of all age groups worldwide continues to rise. It is associated with many other diseases, such as hypertension, stroke, osteoarthritis, gout, non-insulin-dependent diabetes mellitus, and psychiatric disorders [1,2]. In certain special populations, such as the elderly, obesity may directly or indirectly lead to neurological disorders, such as Alzheimer’s disease [3] and multiple sclerosis [4]. In children and adolescents, obesity may lead to a decline in academic performance (AP) [5]. What is more, overweight and obesity can impair people’s cognitive functions, such as memory [6], attention [7], language [8], and executive function (EF) [9,10]. Some studies have found that individuals with obesity exhibit decreased centrality in the left medial frontal gyrus and lateral occipital cortex of the brain (the medial frontal gyrus being a structure implicated in multiple brain circuits associated with attention, EF, and motor function) [11,12]. Furthermore, this group of people demonstrates prolonged reaction times and reduced accuracy in inhibitory control tasks such as the Flanker and Stroop tasks [13]. It can be inferred that compared to the general population, individuals with obesity exhibit weaker resistance to interference and are more prone to making unhealthy dietary choices [14]. It is evident that EF can influence people’s decisions regarding diet and physical activity [15,16]. Therefore, enhancing the EF of individuals with obesity may potentially be beneficial for their health and well-being [17].

EF refers to a cognitive process responsible for a series of higher-order mental functions [18]. It can be divided into (1) core EFs, including inhibitory control, cognitive flexibility (CF), and working memory (WM), and (2) higher-level functions, including problem-solving, planning, and responding [19]. Lifestyle behavior can significantly influence EFs, such as physical activity. In recent years, several reviews have investigated the impact of physical activity on EFs in various populations, such as healthy children and adolescents [20], overweight children and adolescents [21], or the elderly. These reviews have consistently found that exercise improves EFs in these populations. For children and adolescents, EFs are closely related to their AP, social skills, and emotional regulation [22,23]. In studies targeting obese children and adolescents, researchers have found that a combination of aerobic exercise (AE) and resistance training (RT) over a period of several weeks can significantly enhance the WM and inhibitory control of the subjects [10,16]. Another study involved a 3-week intervention with extended exercise time for this population. The results revealed that while it did not effectively reduce the weight of those, this intervention significantly improved their EFs [24]. In addition to the aforementioned long-term interventions, research has also found that after an acute coordinated physical activity intervention (CPA), obese adolescents exhibited significantly reduced reaction times in the Stroop task, indicating an enhancement in their inhibitory control function [25]. In the elderly population, obesity significantly increases the risk of developing type 2 diabetes, [26] leading to notable alterations in brain structure and a heightened probability of neurodegenerative diseases [27]. Regular AE or RT over a span of six months among elderly obese individuals has been shown to enhance their neurocognitive levels [26,28]. Furthermore, RT may also improve other health outcomes in type 2 diabetes patients, such as insulin action and glucose regulation [29]. Furthermore, increasing exercise time to interrupt sedentary behavior, such as walking for 30 min after prolonged sitting, has also been shown to effectively improve the EF of elderly obese individuals [30].

It can be observed that various exercise interventions aimed at improving EFs in this population include AE, RT, AE combined with RT, as well as prolonged exercise duration. However, there is currently a lack of comparative studies evaluating the effectiveness of these methods. This study aims to conduct a systematic review and network meta-analysis (NMA) to compare the effects of different exercises on EFs in overweight or obese individuals. NMA is an extension of traditional pairwise meta-analysis, with the advantage of simultaneously comparing the effectiveness of multiple interventions for a specific outcome. Even when direct comparisons between two interventions are not available within the network structure, NMA can still calculate indirect comparisons [31]. By integrating both direct and indirect evidence, NMA facilitates the ranking of various interventions’ effectiveness, thereby guiding clinicians and patients in selecting the optimal treatment option [32]. We hope that this research will provide insights and references for improving EFs and health management in this population.

## 2. Materials and Methods

### 2.1. Registration

The completion of this systematic review and NMA adheres to the Preferred Reporting Items for Systematic Reviews and Meta-Analyses (PRISMA) guidelines and the Cochrane Collaboration Handbook. The protocol has been registered on the International Prospective Register of Systematic Reviews (PROSPERO, CRD42024533157).

### 2.2. Search Strategy

All research concerning the impact of exercise interventions on the EFs of overweight or obese individuals was comprehensively gathered from the following databases: PubMed, Web of Science, SPORTDiscus, MEDLINE, and CINAHL. The included articles in English should have been published from January 2000 to February 2024. The retrieval details for each database are presented in Appendix A. The search strategy was formulated following the PICOS principle: (P) population: overweight or obese individuals; (I) intervention: acute or long-term exercise intervention; (C) comparator: no exercise; (O) outcomes: the primary outcomes were EFs, which encompass core functions (e.g., inhibitory control, WM, and CF) and higher-level functions (e.g., responding, planning, and problem-solving). Therefore, the primary outcomes should include at least one of the above indicators. Additionally, given the focus of many exercise intervention studies on AP in obese adolescents, secondary outcomes included AP. (S) Study type: only randomized controlled trials (RCTs) were included.

### 2.3. Eligibility Criteria

Studies meeting the following criteria were included: (1) RCTs with data (mean, standard deviation, sample size) suitable for meta-analysis; (2) participants meeting the criteria for obesity or overweight, defined as follows: for European populations, BMI > 25 kg/m^2^ (overweight) and/or BMI ≥ 30 kg/m^2^ (obese); for Asian populations, BMI ≥ 24 kg/m^2^ (overweight) and/or BMI ≥ 28 kg/m^2^ (obese); no specific age requirements for participants; (3) the control group could be a blank control (no exercise intervention) or another form of exercise, but there must be differences in exercise form between the experimental and control groups, rather than just differences in exercise intensity and frequency; (4) measurement of primary or secondary outcomes before and after the intervention.

Studies were excluded if they met any of the following criteria: (1) engaged in academic misconduct such as duplicate publication; (2) were not RCTs involving human subjects; (3) had control and intervention groups that differ not only in exercise mode (e.g., experimental group undergoing exercise intervention, control group undergoing psychological intervention).

### 2.4. Study Selection and Data Extraction

After the database screening, all articles were imported into Zotero software (version 6.0.30), followed by the removal of duplicate documents. Subsequently, abstracts and titles were read, and this step was completed by two researchers independently. If there was any inconsistency in their results, a third researcher intervened. The data extraction section mainly includes the following for each study: (1) basic information (first author, publication year, region); (2) grouping information of the study subjects, gender, and age; (3) intervention measures, intervention load, frequency (times/week), and duration (minutes/time) and total weeks; (4) outcome assessment instruments and indicators.

### 2.5. Quality Assessment

The literature quality assessment was independently conducted by two investigators. The Jadad scale with a total score of 7 was used to assess the quality of the literature. The main evaluation criteria include the following: (1) Generation of Random Sequence—appropriate: generated using computer-generated random numbers or similar methods (2 points); unclear: randomized trial but method of random allocation not described (1 point); inappropriate: alternating allocation method such as odd/even numbers (0 points). (2) Allocation concealment—appropriate: computer control, sealed opaque envelopes, or other methods preventing clinical staff and participants from predicting allocation sequence (2 points); unclear: only indicating the use of random number tables or other random allocation schemes (1 point); inappropriate: alternating allocation, open random number tables, or any measures that do not prevent predictability of grouping (0 points). (3) Blinding—Appropriate (2 points); unclear: trial stated as blinded but method not described (1 point); inappropriate: not using double-blinding or inappropriate blinding methods (0 points). (4) Withdrawals and dropouts—described number and reasons for withdrawals or dropouts (1 point); did not describe number or reasons for withdrawals or dropouts (0 points).

### 2.6. Data Synthesis and Statistical Analyses

The NMA was conducted using the random-effects model within the framework of frequency statistics in Stata 17.0 software. Because various outcome assessment instruments were used across studies and all outcome indicators were continuous variables, effect analysis indicators were represented by the standardized mean difference (SMD) and a 95% confidence interval (CI). Statistical significance was determined if the 95% CI did not overlap with 0. Network plots were generated using the “network” package in the statistical software. Consistency testing was performed using a node-splitting method in cases where closed loops existed between studies, and inconsistent results were addressed using an inconsistency model. Intervention rankings were determined using the surface under the cumulative ranking curve (SUCRA), with higher values indicating better intervention effects. Publication bias was assessed using funnel plots, a common method in meta-analysis. Additionally, in accordance with the Cochrane Handbook recommendations, reverse scaling was applied to certain outcomes to align them with others when the effect measures in studies were opposite to the main direction.

## 3. Results

### 3.1. Study Selection

The flowchart of literature screening is shown in Figure 1. After searching five electronic databases, a total of 2048 records were obtained. After removing 1402 duplicate records, titles and abstracts of 646 studies were screened. Subsequently, full-text reading was conducted for the remaining 48 articles. During this process, one additional eligible study was identified from the reference lists of these articles. Ultimately, 20 studies were included in the meta-analysis.

### 3.2. Study Characteristics

This meta-analysis synthesizes findings from 20 studies published between 2010 and 2023, encompassing a total of 1183 overweight or obese participants (Table 1). The citation format for relevant research can be found in Appendix A. Based on the exercise interventions employed across all studies, interventions were categorized into seven types: (1) control training (CT), involving no exercise intervention, serving as a blank control group; (2) AE, comprising sustained aerobic activities such as running and cycling [17,33,34,35,36,37,38,39]; (3) RT, involving resistance exercises using body weight or external equipment [26,40]; (3) CPA, incorporating multi-body-part movements and/or interaction with objects, requiring precise timing and spatial estimation for goal-directed behavior, with included studies focusing mainly on rope skipping among children and adolescents [25,41,42,43]; (4) prolonged time of exercise (PTE), aimed at increasing participants’ physical activity levels without specifying the type of exercise [17,24,30,44]; (5) high-intensity interval training (HIIT), involving repeated bouts of high-intensity training interspersed with active or passive recovery [17,34,37]; (6) AE combined with RT (mix mode, MIX) [45,46]. Most interventions lasted for at least 4 weeks, although three studies implemented a single session of exercise intervention [25,30,34]. Regarding the study population, approximately 522 (45%) were male participants, with 1 study exclusively including male subjects and 3 studies exclusively including female subjects, while the remaining 16 studies included both male and female participants. Among the included studies, nine focused on children and adolescents aged 18 and under, seven studies targeted adult populations aged 18 to 59, and four studies were conducted on individuals aged 60 and above. Outcome measures across all studies primarily focused on three indicators of EFs: CF, inhibitory control, and WM. Additionally, due to the association between EF and AP in the specific population of children and adolescents, AP was also included in the analysis.

The quality assessment results of the literature are presented in Appendix A. Overall, the quality of the included literature was relatively high, with an average score of 4.75 points. In the context of random allocation, 12 articles implemented appropriate random assignment, with 4 articles also employing allocation concealment. Only five studies conducted double-blind procedures. Achieving complete double-blinding in intervention experiments involving humans is challenging due to the necessity of obtaining informed consent from participants in most studies. Finally, regarding withdrawals and dropout records, 17 articles provided explanations and the number of participants lost to follow-up. The funnel plot graphics of EFs are shown in Appendix A.

### 3.3. Network Meta-Analysis

In the network evidence plot (Figure 2), the size of each node is positively correlated with the sample size involved in each intervention. The thickness of the lines connecting each node represents the number of studies using the respective pair of exercise modalities. AE is the most common intervention, while the combination of AE+RT (MIX) is the least common. Studies using inhibitory control as the outcome measure involve a greater variety of exercises, resulting in better connectivity among the nodes. Table 2 presents the pairwise comparison results of various interventions for different executive functions.

#### 3.3.1. Cognitive Function (CF)

A total of seven studies reported on CF, involving four types of exercise interventions, including AE, CPA, PTE, and MIX. Six studies reported accuracy and reaction time. Figure 3 illustrates the net league table of CF. In CF–accuracy results, significant differences were observed in the effects of the four types of exercise compared to CT. As there was no closed loop for each intervention mode under this indicator, no inconsistency test was required. The SUCRA results showed CPA (SUCRA = 99.2%) > PTE (SUCRA = 75.8%) > MIX (SUCRA = 49.9.0%) > AE (SUCRA = 13.1%) >CT (SUCRA = 11.9%). As for CF-RT, the SUCRA indicated CPA (SUCRA = 96.6%) > PTE (SUCRA = 78.2%) > CT (SUCRA = 44.2%) > AE (SUCRA = 30.9%) > MIX (SUCRA = 0.0%).

#### 3.3.2. Inhibitory Control

A total of 15 articles reported on inhibitory control, involving five exercise modalities: AE, RT, PTE, HIIT, and MIX. Seven articles reported accuracy, while twelve articles reported reaction time. Node-splitting analysis revealed no significant inconsistency among different types of exercise (*p* > 0.05), indicating high reliability of the results. The SUCRA results showed the following: HIIT (SUCRA = 63.0%) > PTE (SUCRA = 58.0%) > RT (SUCRA = 55.4%) > AE (SUCRA = 38.4%) > CT (SUCRA = 35.2%). Regarding reaction time, node-splitting analysis revealed inconsistencies between HIIT and CT as well as AE (*p* < 0.05). CPA, PTE, HIIT, and MIX showed significant differences compared to CT. And significant differences were observed between CPA and MIX, HIIT, and PTE. The SUCRA results indicated CPA (SUCRA = 89.3%) >HIIT (SUCRA = 63.6%) > RT (SUCRA = 50.2%) > PTE (SUCRA = 41.8%) > MIX (SUCRA = 39.9%) > AE (SUCRA = 39.3%) > CT (SUCRA = 26.0%).

Five articles reported on WM, involving three types of exercise interventions: AE, RT, and MIX. Among these, five articles reported accuracy, while two reported reaction time. As there was no closed loop for each intervention mode under this indicator, no inconsistency test was required. The SUCRA results showed RT (SUCRA = 94.8%) > CT (SUCRA = 50.2%) > MIX (SUCRA = 31.2%) > AE (SUCRA = 23.8%). Studies on reaction time only involve aerobic exercise; there is no SUCRA.

#### 3.3.3. Academic Performance (AP)

Four articles reported on AP, involving two types of exercise interventions: CPA and MIX. Figure 3 indicates significant differences between CPA, MIX, and CT, as well as between MIX and AE. The SUCRA results showed AE (SUCRA = 100.0%) > CT (SUCRA = 66.6%) > CPA (SUCRA = 33.0%) > MIX (SUCRA = 0.4%).

#### 3.3.4. Subgroup Analysis

The EF results were subjected to subgroup analysis based on age and intervention duration, with detailed results provided in Figure 3. HIIT intervention only involved acute intervention and adolescent groups, while MIX only involved adolescents and interventions lasting over 16 weeks, thus not requiring subgroup analysis. AE showed a significant difference compared to CT (SMD = 0.27, 95%CI = 0.02~0.53, *p* = 0.036). In the age subgroup analysis, AE exhibited a significant effect on interventions for the 0–17 age group (SMD = 0.84, 95%CI = 0.31~1.38, *p* = 0.002), while in the intervention duration subgroup analysis, interventions lasting 8–16 weeks showed significant improvement in EFs (SMD = 0.53, 95%CI = 0.00~1.05, *p* = 0.048). What is more, in children and adolescents, subgroup analysis results revealed significant differences for acute (SMD = 1.12, 95%CI = 0.45~1.80, *p* = 0.001) and 0–8 week (SMD = 0.71, 95%CI = 0.27~1.15, *p* = 0.001) CPA intervention compared to CT. Additionally, PTE showed significant effects for individuals aged 17–59 (SMD = 0.93, 95%CI = 0.11~1.75, *p* = 0.027) and had marginally significant effects on improving EFs in adults aged 59 and older (SMD = 0.62, 95%CI = −0.02~1.25, *p* = 0.058).

## 4. Discussion

This systematic review and NMA included 20 RCTs, comprising a total of 1183 obese and overweight participants. It aimed to compare the effects of different exercise modalities on CF, inhibitory control, WM, and adolescents’ AP. The results revealed that CPA and AE have significant benefits for CF, inhibitory control, and AP in children and adolescents. Furthermore, PTE improves EFs in adults and older adults.

### 4.1. Exercise Effect on EFs

In previous similar meta-analyses, exercise interventions have consistently been found to significantly improve EFs in elderly individuals, obese adolescents, individuals with mild cognitive impairment, and healthy populations [20,47,48,49]. The exercise interventions in these studies are diverse, involving long-term exercise (6 weeks or longer) [20], after-school exercise for children or adolescents [47], and combined exercise, among others [49]. Exercise not only improves individuals’ EFs directly, but it also activates brain regions associated with EFs, including the anterior cingulate and superior frontal gyrus. Additionally, exercise indirectly enhances EFs by improving physical fitness components such as speed, agility, and cardiorespiratory fitness [15]. However, there are currently no studies comparing the effects of different exercise modalities on EFs in obese populations.

In the results, we found that CPA has a significantly positive effect on children and adolescents’ CF and inhibitory control. The CPA interventions included in this study involved jump rope exercises. Previous researchers have found that after CPA intervention, inhibitory control improves in healthy children, children with ADHD, and overweight/obese children and adolescents [50]. Additionally, it can enhance the performance of obese individuals facing food cue-related task conditions of the Stroop task [25,51]. Some studies suggest that children and adolescents, when engaged in more complex coordinated movements, first activate the cerebellum and prefrontal cortex, thereby enhancing subsequent cognitive performance [52]. Furthermore, CPA has been found to improve multiple physical fitness components, such as strength, aerobic capacity, and flexibility, all of which are correlated with BMI [51]. It can be observed that, compared to conventional exercises such as AE, CPA not only improves BMI and cardiorespiratory fitness but also enhances coordination. This complex and coordinated activity pre-activates the prefrontal cortex and cerebellum, thereby improving subsequent cognitive performance [52,53]. However, as the three studies involving CPA included in the research were all conducted on children and adolescents, we cannot directly determine the impact of this exercise on adults or the elderly. Some studies have also found that in older adults performing coordinated and flexible movements, there is greater activation in sensorimotor and visual–spatial networks [54]. In the future, it is hoped that more RCT studies will explore the impact of CPA on the cognition of adults or the elderly.

Additionally, PTE also demonstrates positive effects on improving adults’ and old people’s CF and inhibitory control. It may regulate leptin, cortisol, and brain-derived neurotrophic factor levels, as well as enhance neural synapses and potentiation, which could have favorable effects on reaction times among obese adults [24,55]. Studies have found that prolonged uninterrupted sitting behavior can increase fatigue levels in individuals [56], leading to functional alterations in the nervous and endocrine systems, such as impaired autonomic nervous system function, particularly in chronically fatigued individuals [57,58]. Moreover, prolonged uninterrupted sitting can elevate blood glucose levels and promote the occurrence of insulin resistance. This metabolic disruption can have negative effects on glucose metabolism in the brain, potentially leading to hippocampal atrophy and harmful effects on the density of the medial temporal lobe, posterior cingulate cortex, and precuneus. This may be due to enhanced neuronal activity, reduced cerebral blood flow, and decreased synaptic plasticity. The reduction in cortical density is believed to be a cause of early cognitive decline and dementia risk [55,59]. In a study by Wheeler et al. (2020) focusing on obese, sedentary older adults, it was found that breaking sedentary behavior with physical activity could increase the subjects’ serum brain-derived neurotrophic growth factor levels as well as improve WM [60]. PTE, such as increasing daily walking steps and breaking prolonged sitting behavior, may improve cortical circulation, muscle tone, and readiness. Even though such activities may have lower intensity compared to AE or RT, they may be more effective in improving inhibitory control in OW/OB by interrupting prolonged sedentary behavior, compared to adding a concentrated period of MVPA [61].

We can also observe that RT and AE have effective roles in improving WM and AP in obese populations. Vints et al. (2024) conducted a 12-week RT intervention on elderly individuals with mild cognitive impairment. The results revealed that the increase in Insulin-like Growth Factor-1 levels was associated with improved reaction times in mathematical processing. Additionally, the elevation in serum Interleukin-6 (IL-6) levels was associated with improvements in memory search reaction speed [62]. IL-6, also known as “exercise factor,” is associated with inflammatory responses. Its release promotes the formation of anti-inflammatory cytokines, such as IL-1RA, IL-10, and corticosteroids [63,64]. Obese individuals exhibit a state of “low-grade chronic inflammation” [65], characterized by increased levels of pro-inflammatory cytokines in both the circulatory system and adipose tissue. IL-6, as a pro-inflammatory cytokine, can influence neuroinflammatory responses in the brain, thereby impairing cognitive function [66]. At the molecular level, the connection between IL-6 and memory function has been previously explained through cytokine involvement in synaptic formation, neurogenesis, and memory consolidation [67]. Due to the highest expression of inflammatory cytokine receptors for IL-6 in the hippocampus, changes in peripheral IL-6 levels may affect hippocampus-related memory scores [68]. Consistent with previous research findings, after a 10-week aerobic exercise intervention, children’s dual-task performance significantly improved [69]. This study also found that AE has a positive effect on improving AP, which may be attributed to AE’s ability to enhance cardiorespiratory fitness. A longitudinal analysis spanning 2 years, involving 1802 youths, found that after adjusting for other physical health components and BMI, maintaining higher levels of cardiorespiratory fitness and exercise capacity is associated with higher AP [70]. This could be because higher cardiorespiratory fitness is positively correlated with gray matter volume in the brain, specifically in frontal regions (e.g., supplementary motor cortex and premotor cortex), subcortical regions (e.g., hippocampus), temporal regions (e.g., inferior temporal gyrus and parahippocampal gyrus), and the calcarine cortex. The premotor cortex, supplementary motor cortex, and hippocampus were associated with better AP [71]. Real et al. (2015) also found that AE increases the presynaptic protein and postsynaptic GluA1 and GluA2/3 receptors in the motor cortex, thereby enhancing synaptic efficiency. This may further elevate children’s cognitive levels and AP [72].

### 4.2. Age and Intervention Duration

In this study, subgroup analysis based on age and intervention duration revealed significant improvement in EFs with AE and CPA interventions for children and adolescents aged 0–17. Combining these findings with the results related to AP, it appears that children and adolescents may benefit more from AE. The significant effect of AE interventions lasting 8–16 weeks is consistent with previous research, suggesting that chronic exercise can promote higher-order cognitive functions [73]. However, some studies suggest that acute aerobic exercise may have a greater impact on EFs [74]. This discrepancy could be related to differences in the physiological characteristics of the study participants. The former study focused on late-middle-aged healthy adults, while our study focused on overweight or obese individuals. Long-term exercise may promote adaptive changes in their cardiovascular function, body composition, and nervous system, thereby enhancing their EFs [20,70,75]. In summary, we recommend that OW/OB children and adolescents engage in regular AE over the long term, in addition to increasing their physical activity duration and reducing sedentary behavior.

Additionally, PTE has a significant effect on improving EFs in adults and older individuals, perhaps because this age group is more prone to be sedentary. Technology development in the past few decades has brought about significant changes in the nature of workstations, allowing people to complete various tasks without the need for additional activities, which has exacerbated people’s sedentary behavior during working hours [76,77]. Therefore, it is necessary for adults to change this state. Additionally, by 2050, 20% of the global population is projected to be aged 65 years or older, and the widespread prevalence of sedentary lifestyles in this age group will present significant public health challenges [78]. On one hand, older adults may face chronic health issues that can limit their mobility. Conditions such as arthritis, heart disease, and cognitive decline can significantly reduce their ability to remain active [79]. On the other hand, psychological factors also play a role. Older adults may be reluctant to engage in even low-intensity exercises due to concerns about injury or falls, despite the potential health benefits of these activities [80]. However, inactivity can further diminish inhibitory control, making it essential to reduce and restructure prolonged sedentary behavior, such as interrupting sedentary periods of more than 20 min by standing and walking [60,61]. So, PTEs can improve this issue and are suitable for older adults because they do not focus on exercise intensity [60].

### 4.3. Limitations and Implications for Future Study

This study has several limitations. Firstly, NMA has its own limitations. For example, inconsistencies between direct and indirect comparisons, incomplete data in some studies, and heterogeneity among studies can lead to uncertain results. Secondly, to ensure the reliability of the outcomes, we only included RCTs, which resulted in a small number of studies included in this paper, with only 20 articles. This limitation may further lead to shortcomings in subgroup analysis (such as not grouping by factors like gender, region, BMI, etc.). Furthermore, the duration of long-term interventions ranged from 4 weeks to 20 weeks, making it difficult to precisely align intervention times, which could introduce bias. It also implies that there are few RCTs investigating the effects of exercise interventions on EFs in OW/OB populations, and the variety of exercise intervention methods is limited, such as CPA interventions only involving jump rope exercises. Additionally, literature searches revealed that most studies only include one or a few aspects of inhibitory control, CF, and WM. The use of various neuropsychological tests in different studies may also lead to inconsistent results. In the future, a more comprehensive set of evaluation indicators should be selected for EF in specific populations.

## 5. Conclusions

This systematic review and NMA found that CPA and AE have significant benefits for CF, inhibitory control, and AP in children and adolescents. Furthermore, PTE improves EFs in adults and older adults. Overall, based on the results of this study and previous related research, we suggest that OW/OB individuals should start by interrupting prolonged sedentary behavior and increasing fragmented physical activity, and gradually incorporate AE, RT, and CPA (such as skipping rope) into physical activity time, especially children and adolescents.

## Figures and Tables

**Figure 1 behavsci-14-01227-f001:**
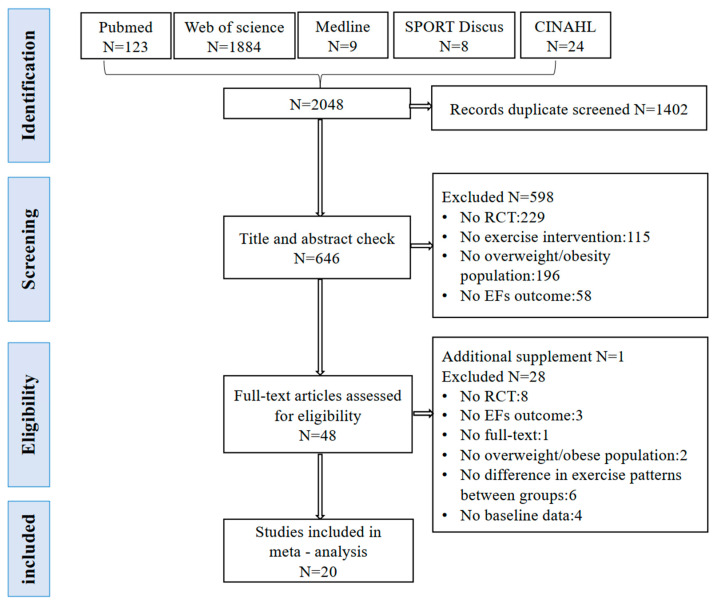
The flowchart of systematic review and meta-analysis (PRISMA) depicting the study selection process.

**Figure 2 behavsci-14-01227-f002:**
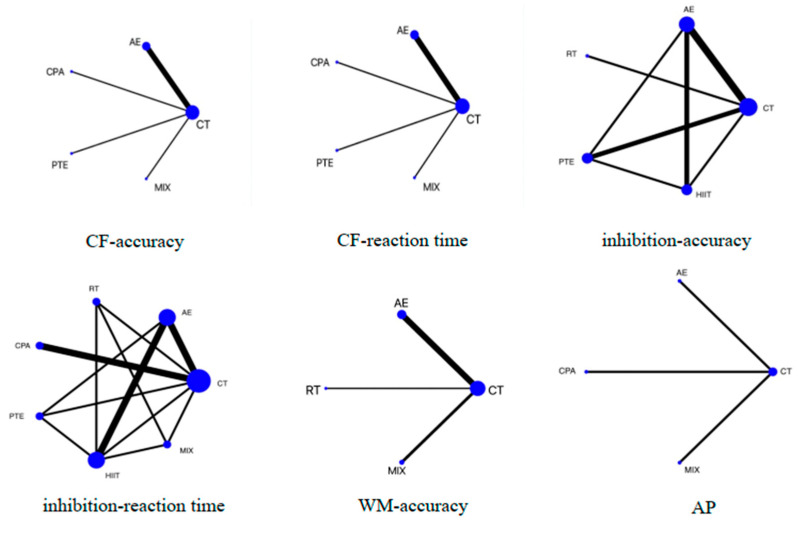
Network plot presenting the effects of different exercise types on executive functions (EFs) and academic performance. CF, cognitive flexibility; WM, working memory; AP, academic performance; CT, control training; AE, aerobic exercise; RT, resistance training; PTE, prolonged time of exercise; MIX: AE+RT.

**Figure 3 behavsci-14-01227-f003:**
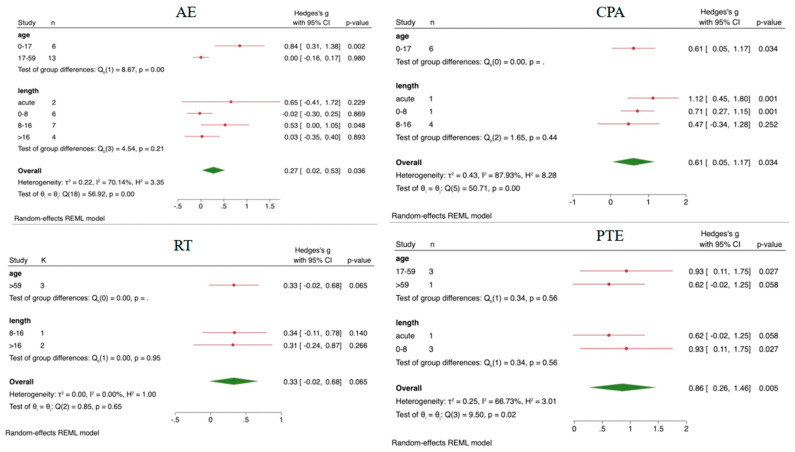
The results of subgroup analysis. AE, aerobic exercise; RT, resistance training; PTE, prolonged time of exercise; CPA, coordinated physical activity; Red line, the effect value of a single item; Green symbols, over all effect value.

**Table 1 behavsci-14-01227-t001:** Characteristics of included studies.

	Study; Country/Region	Participant Characteristics			Intervention					Outcome Measures
		Sample	Participants (Age Range; Sex-M%)	Sample Size (IG/CG)	Design	Type	Intensity	Frequency	Length (mins)	
1	Abel et al., 2023; Spain	OW/OB	10.03 ± 1.51; 59%	101 (51/50)	AE+RT (60 min of AE and 30 min of RT)	Chronic	80% of the maximal heart rate and at the level of the anaerobic threshold	90 min/session, 3–5 × wk	20 wk (9000)	1. CF:Design Fluency Test and Trail Making Test; 2. Inhibition:modified version of the Stroop test; 3. WM:Delayed Nonmatch-to-Sample (DNMS) computerized task; 4. AP:Spanish version of the Woodcock-Johnson III Tests of Achievement
2	Furlano et al., 2023; Canada	OW/OB	68.7 ± 5.7; 50%	24 (13/11)	RT (2 sets of 6 to 8 repetitions of exercises)	Chronic	80% × 1RM	60 min/session, 3 × wk	24 wk (1440)	1. inhibition:Stroop Test (condition C–B); 2. WM:Digit Span Test;
3	Domal et al., 2023; India	OW/OB	24 ± 4.5 (65%)	13 (7/6)	PTE (individualized graded target for each week via step count)	Chronic	-	-	8 wk (-)	1. inhibition:Eriksen flanker test
4	Chou et al., 2023; China	OW	10.0 ± 1.1 (58.7%)	50 (25/25)	AE (competitive team games, i.e., running, jump rope)	Chronic	MVPA	40 min/session, 5 × wk	10 wk (2000)	1. inbibition:Stroop Test of the Vienna Test System; 2. AP:Taiwan’s Ministry of Education was used to measure competence in Chinese language and mathematics in fifth and sixth graders.
5	Oliveira et al., 2022; Brazil	OW/OB	31.3 ± 7.1 (40.6%)	64 (20/21/23)	HIIT	Chronic	VPA	30 min/session, 3 × wk	12 wk (1080)	1. inhibition:the Stroop Color-Words Test
					AE		MVPA			
					PTE		-			
6	Zhang et al., 2022, China	OW/OB	11.56 ± 1.03 (79.2%)	57 (19/19/19)	HIIT	Acute	VPA	30 min/session	-	1. Inhibition: the Stroop Color and Word Test.
					AE		VPA			
					CT		-			
7	Zlibinaite et al., 2021, Lithuania	OW/OB	46.8 ± 6.2 (0%)	33 (17/16)	AE (cycling sessions)	Chronic	MPA	60 min/session, 5 × wk	8 wk (2400)	1. Inhibition: the Stroop Color and Word Test: go/no-go task; 2. WM: mathematical processing task; 3. CF: 2-choice reaction time task.
8	Zlibinaite et al., 2020, Lithuania	OW/OB	44.9 ± 6.2 (0%)	26 (13/13)	AE	Chronic	MPA	50 min/session, 3 × wk	24 wk (3600)	1. WM: mathematical processing task; 2. CF: 2-choice reaction time task; 3. Memory: verbal working memory.
9	Zhang et al., 2020, China	OB	14.6 ± 0.7 (47.4%)	38 (19/19)	CPA (jump rope)	Acute	MPA	-	30 min	1. Inhibition: modified food-cue-related Stroop test.
10	Chou et al., 2020, China	OW	12.19 ± 0.68 (61.9%)	84 (44/40)	CPA (movement game)	Chronic	-	40 min/session, 3 × wk	8 wk (960)	1. Inhibition: Stroop test.
11	Inoue et al., 2020	OB	30.0 ± 5.4 (100%)	20 (10/10)	HIIT/AE	Chronic	VPA	40 min/session, 3 × wk	6 wk (720)	1. Inhibition: Stroop Color and Word Test.
12	Liu et al., 2018, China	OB	14.06 ± 0.83 (57.1%)	70 (35/35)	CPA (jump rope)	Chronic	VPA	75 min/session, 2 × wk	12 wk (1800)	1. Inhibition: classic Stroop color–word conflict task/food-cue condition.
13	Allom et al., 2018, Australia	OB	41.39 ± 7.85 (13.75%)	80 (42/38)	PTE	Chronic	-	-	5 wk	1. CF: Wisconsin Card Sorting Test/paper and pencil Trail Making Test.
14	Quintero Gacharná et al., 2018	OW/OB	44.38 ± 8.59 (21.6%)	95 (50/45)	RT/HIIT/RT+HIIT	Acute	MPA	-		1. Inhibition: Stroop Color and Word Test.
15	Chen et al., 2017, China	OB	14.05 ± 0.83 (57.6%)	66 (33/33)	CPA (jump rope)	Chronic	MVPA	120 min/session, 6 × wk	12 wk (8640)	1. CF: Tower of London—Drexel task.
16	Wennberg et al., 2015, Australia	OW/OB	59.7 ± 8.1 (52.6%)	38 (19/19)	PTE	Acute	-	-	-	1. Inhibition: modified Stroop color–word task.
17	Chen et al., 2016, China	OB	12.74 ± 0.73 (56%)	50 (25/25)	AE	Chronic	MVPA	40 min/session, 4 × wk	12 wk (1920)	1. CF: Wisconsin Card Sorting Test.
18	Dao et al., 2013, Canada	OW	69.44 ± 2.91 (0%)	77 (41/36)	RT	Chronic	MVPA	40 min/session, 2 × wk	13 wk (3840)	1. Inhibition: the Stroop test.
19	Davis et al., 2007, USA	OW	9.3 ± 1.0 (44%)	116 (56/60)	CPA (jump rope)	Chronic	MVPA	40 min/session, 7 × wk	16 wk (1080)	1. AP: Woodcock–Johnson Tests of Achievement III.
20	Smith et al., 2010, USA	OW/OB	52.3 ± 9.6 (36%)	81 (43/38)	AE	Chronic	MVPA	40 min/session, 5 × wk	12 wk (3641)	1. Inhibition: Stroop Interference; 2. CF: Trail Making Test B-A; 3. WM: Digit span.

OW, overweight; OB, obese; AE, aerobic exercise; RT, resistance training; PTE, prolonged time of exercise; AE+RT, the combination of aerobic exercise and resistance training; CPA, coordinated physical activity; MVPA, moderate-to-vigorous-intensity physical activity; VPA, vigorous-intensity physical activity; CF, cognitive flexibility; WM, working memory; AP, academic performance; IG, intervention group; CG, control group.

**Table 2 behavsci-14-01227-t002:** Net league table with network evidence.

	CT	2.72 (2.35, 3.09)	7.15 (5.03, 9.27)	12.79 (7.55, 18.03)	0.04 (−1.33, 1.41)			CF–accuracy
CF-RT	13.65 (10.31, 16.99)	MIX	4.43 (2.28, 6.58)	10.07 (4.82, 15.32)	−2.68 (−4.10, −1.26)			
	−20.54 (−33.86, −7.22)	−34.19 (−47.92, −20.46)	PTE	5.64 (−0.01, 11.29)	−7.11 (−9.64, −4.59)			
	−39.12 (−68.02, −10.22)	−52.77 (−81.86, −23.68)	−18.58 (−50.40, 13.24)	CPA	−12.75 (−18.17, −7.33)			
	0.02 (−0.03, 0.07)	−13.63 (−16.97, −10.29)	20.56 (7.24, 33.88)	39.14 (10.24, 68.04)	AE			
	CT	0.08 (−0.17, 0.33)	0.06 (−0.17, 0.29)	1.40 (−15.66, 18.46)	0.02 (−0.23, 0.27)			inhibition–accuracy
inhibition–RT	−2.19 (−6.20, 1.82)	HIIT	−0.02 (−0.14, 0.11)	1.32 (−15.74, 18.38)	−0.06 (−0.21, 0.09)			
	−0.61 (−6.60, 5.37)	1.58 (−3.95, 7.11)	PTE	1.34 (−15.72, 18.40)	−0.04 (−0.17, 0.08)			
	−1.52 (−7.37, 4.33)	0.68 (−5.40, 6.75)	−0.91 (−8.78, 6.96)	RT	−1.38 (−18.44, 15.68)			
	−0.74 (−3.93, 2.45)	1.45 (−2.23, 5.12)	−0.13 (−5.55, 5.29)	0.77 (−5.49, 7.04)	AE			
	0.75 (−4.29, 5.80	1.44 (−4.34, 7.22)	−0.14 (−7.56, 7.28)	4.40 (−2.23, 11.03)	0.76 (−5.47, 7.00)	MIX		
	−5.15 (−9.37, −0.93)	−2.96 (−8.70, 2.79)	−4.54 (−12.09, 3.01)	−3.64 (−10.77, 3.50)	−4.41 (−9.73, 0.91)	−4.40 (−11.03, 2.23)	CPA	
	CT	−0.30 (−1.52, 0.92)	2.20 (−0.72, 5.12)	−0.50 (−1.96, 0.97)	WM–accuracy	WM-RT	CT	
AP	CT	MIX	2.50 (−0.66, 5.66)	−0.20 (−2.11, 1.71)			0.01 (−0.35, 0.32)	AE
	3.54 (3.12, 3.96)	MIX	RT	−2.70 (−5.96, 0.57)				
	2.00 (0.89, 3.11)	−1.54 (−2.72, −0.36)	CPA	AE				
	−2.88 (−4.27, −1.49)	−6.42 (−7.87, −4.97)	−4.88 (−6.66, −3.10)	AE				

AE, aerobic exercise; RT, resistance training; PTE, prolonged time of exercise; MIX, the combination of aerobic exercise and resistance training; CPA, coordinated physical activity; CF, cognitive flexibility; WM, working memory; AP, academic performance; WM, working memory.

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
