# Peer review of "The Impact of Different Types of Exercise on Executive Functions in Overweight/Obese Individuals: A Systematic Review and Network Meta-Analysis"

_behavsci, 2024, doi:10.3390/bs14121227_

Round 1
Reviewer 1 Report
Comments and Suggestions for Authors
1) In the section mentioning the ‘exercise factor’ (365 line) IL-6 and its relation to memory functions, it might be useful to include an additional discussion on the specific effects of IL-6 in obese or overweight subjects, as these might differ from non-obese populations.
2) The discussion includes a detailed analysis of the differences in the impact of each intervention. However, in the analysis of CPA and PTE, it may be useful to explore further why prolonged coordinated exercise has superior effects on some executive functions compared to other modalities.
3) Academic performance is a term that encompasses the final grade of a teaching-learning process, which can be influenced by the method used by the teacher, the subject assessed and even the grading system used. Therefore, it may be interesting to expand the rationale between improvements in academic performance and specific components of executive function, in order to better understand how cognitive changes can be translated into academic skills.
4) The manuscript could benefit from a concise summary that synthesises the practical implications and key recommendations at the end of the discussion, guiding practitioners and academics on how to implement the findings for the improvement of executive function in obese populations.
Author Response
Comment 1:In the section mentioning the ‘exercise factor’ (365 line) IL-6 and its relation to memory functions, it might be useful to include an additional discussion on the specific effects of IL-6 in obese or overweight subjects, as these might differ from non-obese populations.
Respond 1: Thank you for your professional comments. Based on your suggestions, I have added more detailed information about IL-6 and cognitive function in lines 382-385: "Obese individuals exhibit a state of 'low-grade chronic inflammation,' characterized by increased levels of pro-inflammatory cytokines in both the circulatory system and adipose tissue. IL-6, as a pro-inflammatory cytokine, can influence neuroinflammatory responses in the brain, thereby impairing cognitive function."
Comment 2:The discussion includes a detailed analysis of the differences in the impact of each intervention. However, in the analysis of CPA and PTE, it may be useful to explore further why prolonged coordinated exercise has superior effects on some executive functions compared to other modalities.
Respond 2: Thank you for your professional and detailed advice. Based on your suggestions, I have made two additions in the original text.
(1)In line 343-352:It can be observed that compared to conventional exercises such as AE, CPA not only improves BMI and cardiorespiratory fitness but also enhances coordination. This complex and coordinated activity pre-activates the prefrontal cortex and cerebellum, thereby improving subsequent cognitive performance. 52,53 However, as the three studies involving CPA included in the research were all conducted on children and adolescents, we cannot directly determine the impact of this exercise on adults or the elderly. Some studies have also found that in older adults performing coordinated and flexible movements, there is greater activation in sensorimotor and visual-spatial networks.54 In the future, it is hoped that more RCT studies will explore the impact of CPA on the cognition of adults or the elderly.
(2)In line 354-356:Additionally, PTE also demonstrates positive effects on improving adults and old people’s CF and inhibitory control. It may regulate leptin, cortisol, and brain-derived neurotrophic factor levels, as well as enhance neural synapses and potentiation, which could have favorable effects on reaction times among obese adults. 24,55
Comment 3: Academic performance is a term that encompasses the final grade of a teaching-learning process, which can be influenced by the method used by the teacher, the subject assessed and even the grading system used. Therefore, it may be interesting to expand the rationale between improvements in academic performance and specific components of executive function, in order to better understand how cognitive changes can be translated into academic skills.
Respond 3: We greatly appreciate your professional and in-depth suggestions. Indeed, the evaluation of academic performance (AP) is influenced by many factors, including teachers' assessment criteria and the tools used. Different aspects of executive function can impact AP from various angles. Your suggestions are highly insightful and point to a topic well worth further research. However, the primary focus of this study is on the effects of exercise on executive function (including its impact on academic performance). Therefore, we have not delved into how executive function influences AP in this paper.
Your suggestion is indeed excellent and addresses a topic that could significantly enhance our understanding of the relationship between cognitive function and academic performance. We plan to explore this area in future research to better help people understand this connection. Once again, thank you for your insightful and thought-provoking suggestions.
Comment 4: The manuscript could benefit from a concise summary that synthesises the practical implications and key recommendations at the end of the discussion, guiding practitioners and academics on how to implement the findings for the improvement of executive function in obese populations.
Respond 4: Thank you for your professional suggestions. Your recommendations have helped enhance the guiding role of our research. Accordingly, we have made revisions to the conclusion section: This systematic review and NMA found that CPA and AE have significant benefits for CF, inhibitory control, and AP in children and adolescents. Besides, PTE improves EFs in adults and older adults. Overall, based on the results of this study and previous related research, we suggest that OW/OB should start by interrupting prolonged sedentary behavior and increasing fragmented physical activity, and gradually incorporate AE, RT, and CPA (such as skipping rope) into physical activity time, especially for children and adolescents.
Reviewer 2 Report
Comments and Suggestions for Authors
Overview and general recommendation:
The present article “The impact of different types of exercise on executive functions 2 in overweight/obese individuals: A Systematic Review and 3 Network Meta-Analysis” proposes a network meta-analysis to compare the effects of different exercise training on executive function (EF) in obese or overweight individuals. The authors included findings from a total of 20 RCTs. They categorized the types of intervention into seven (i.e., Control training-CT, Aerobic exercise-AE, resistance training-RT, Coordinated physical activity-CPA, Prolonged time of exercise-PTE, High-intensity interval training-HIIT, AE combines RT-mix mode, MIX). The main results demonstrated the preferable effects of various interventions on EF improvement using the surface under the cumulative ranking curve (SUCRA). Furthermore, the authors performed a subgroup analysis based on age and intervention duration. The primary findings showed that PTE and CPA had beneficial effects on improving CF and inhibitory control in obese populations, while RT and AE improved WM and adolescents’ AP. When focusing on the subgroup analysis based on age and intervention duration, AE and PTE were found to significantly enhance EFs in the 0-17 age group, and PTE had a significant effect on EFs in individuals aged 59 and above. Additionally, AE lasting 8-16 weeks and acute CPA interventions significantly improved EFs.
Overall, this well-written and high-quality study introduces a novel statistical metric that provides a way to present the ranking of treatments in an interpretable way and helps communicate the results of a complex analysis. Nevertheless, there are some adjustments the authors need to make in order to publish this manuscript.
Major comments
1. The authors must review the final bibliography since they have included multiple articles (i.e., Abel et al., Mora-Gonzalez et al., and Ortega et al.) from the same project, Activebrains. However, the authors reported different design interventions, intensities, and frequencies. Finally, the results of Abel et al. (2023) included the same cognitive tests as Ortega et al. (2022) but presented the participants' genetic profiles when interpreting the results. Kindly verify that this situation does not occur with other studies in the final analysis.
2. The authors must include more detailed information about the NMA approach in the proper section.
3. Overall, comprehending the reported results presents a challenge because additional information is necessary to account for various exercise modalities with differing durations across distinct populations, such as children and older adults.
4. Given that the authors incorporated various intervention types and diverse populations, including children, young adults, and older adults, it is essential for them to enhance the discussion section by providing additional insights on the clinical relevance and practical implications of these findings.
Minor comments:
Ln 71 – Responding was mentioned as “reasoning” in the abstract. Please ensure consistency throughout the document.
Ln 71 – When presenting physical activity, it makes more sense to talk about “lifestyle behaviors” rather than “environmental factors.”
Introduction section – Please add more details when presenting the different exercise interventions through this section. How were these interventions defined in the articles cited?
Ln 89 – What the authors mean when mentioning “neurocognitive levels.” Can you please clarify it?
Ln 91-92 – The authors should define the other types of exercise interventions previously presented.
Material and methods – Why didn’t the authors add a minimum intervention duration as an inclusion criterion? How can we compare 4-week interventions with 20-week interventions?
To determine which exercise intervention yields the most significant benefits for enhancing executive function, it’s essential to align the types of interventions closely. Avoid comparing a 2-week intervention to a 10-week one; if such comparisons are made, they should be detailed in the limitations section and thoroughly discussed.
Ln 201-202 - how can you compare acute vs. longitudinal intervention effects and, at the same time, different types of interventions? Can you please clarify how the statistical approach accounts for all of this?
Ln 230 - This section needs more detail. Could you clarify how these lines illustrate the effects? Do they correlate with specific numbers? Additional information would be helpful.
Ln 237-238 – Please add the rest of the acronyms in Figure 2 (e.g., CT).
Results section – Understanding the authors' presentation of the Surface Under the Cumulative Ranking (SUCRA) for exercise interventions across various populations poses significant challenges. Should we anticipate that all types of exercise interventions exert identical effects on all demographic groups, such as children versus adults?
This represents a presumptive assumption, as the variability observed among the results is contingent upon the types of interventions and the populations involved.
Ln 254 – While the text refers to it as “table 3”, line 266 states “figure 3”. Also, more detailed information on the reported values is required.
Discussion section – The authors should expand this section in order to properly understand how the NMA analysis may explain the differences in the effects they present in the text.
Should we expect the same effects on cognition due to the same exercise intervention in children vs. adults with obesity? What role do the various phases of neuroplasticity play in these effects? Should we include all these interventions without taking the specific population into account?
Discussion section—When discussing the subgroup analysis, the authors did not address why the results focus on young or older populations. Are there distinct implications of being overweight or obese in childhood compared to adulthood?
Limitations section—Please include the limitations the NMA presents per se (e.g., it involves complex statistical models and requires careful handling of study quality, data heterogeneity, and methodological issues). How did the authors handle the heterogeneity of the duration and populations?
Ultimately, the authors' analysis of the effects of EFs involves various neuropsychological tests used in each study, which should be recognized as a limitation.
Author Response
We greatly appreciate your professional and meticulous feedback. We are truly touched by the thoroughness of your work, which has provided us with numerous valuable suggestions. Based on your recommendations, we have made revisions accordingly. For detailed changes, please see the attached Word document.

Reviewer 3 Report
Comments and Suggestions for Authors
This meta-analysis strongly supports the idea that physical activity positively influences the executive functions of overweight and obese individuals, with targeted interventions capable of enhancing specific aspects of these cognitive functions.
While the study is well-designed and contributes valuable insights, some areas—such as follow-up duration and sample diversity—could be refined to give a more complete understanding of exercise effects across a broader population. Nonetheless, the findings offer a solid foundation for future studies and for introducing exercise programs aimed at individuals with obesity.
Although 8–16-week interventions show notable benefits, there is limited discussion on whether extending the duration might yield further improvements. Additionally, it is unclear if the positive outcomes observed in participants aged 59 and older would apply to other older adults with specific health profiles.
Below are references to sources covering different aspects of this topic. We believe that including these could add depth to the analysis. We would be grateful if you could review them, consider their inclusion, and make any necessary adjustments:
C. Cadenas-Sanchez et al. "An exercise-based randomized controlled trial on brain, cognition, physical health and mental health in overweight/obese children (ActiveBrains project): Rationale, design and methods.." Contemporary clinical trials, 47 (2016): 315-24 . https://doi.org/10.1016/j.cct.2016.02.007.
C. Davis et al. "Effects of aerobic exercise on overweight children's cognitive functioning: a randomized controlled trial.." Research quarterly for exercise and sport, 78 5 (2006): 510-9 . https://doi.org/10.1249/00005768-200605001-01007.
G. O'Malley et al. "Aerobic exercise enhances executive function and academic achievement in sedentary, overweight children aged 7-11 years.." Journal of physiotherapy, 57 4 (2011): 255 . https://doi.org/10.1016/S1836-9553(11)70056-X.
J. Mora-Gonzalez et al. "Physical Fitness, Physical Activity, and the Executive Function in Children with Overweight and Obesity." The Journal of Pediatrics, 208 (2019): 50–56.e1. https://doi.org/10.1016/j.jpeds.2018.12.028.
Jamie C. Peven et al. "The Effects of a 12-Month Weight Loss Intervention on Cognitive Outcomes in Adults with Overweight and Obesity." Nutrients, 12 (2020). https://doi.org/10.3390/nu12102988.
Marta Maria da Silva Lira Batista et al. "Effects of neuromodulation on executive functions and food desires in individuals with obesity: a systematic review.." Nutricion hospitalaria (2022). https://doi.org/10.20960/nh.04100.
Michael J. Wheeler et al. "Distinct effects of acute exercise and breaks in sitting on working memory and executive function in older adults: a three-arm, randomised cross-over trial to evaluate the effects of exercise with and without breaks in sitting on cognition." British Journal of Sports Medicine, 54 (2019): 776 - 781. https://doi.org/10.1136/bjsports-2018-100168.
Quintero Gacharná et al. "Acute effect of three different exercise training modalities on executive function in overweight inactive men: the brainfit study." Medicine and Science in Sports and Exercise, 50 (2018): 202. https://doi.org/10.1249/01.MSS.0000535750.40951.C1.
S. Ludyga et al. "A network meta-analysis comparing the effects of exercise and cognitive training on executive function in young and middle-aged adults." European Journal of Sport Science, 23 (2022): 1415 - 1425. https://doi.org/10.1080/17461391.2022.2099765.
Xia Xu et al. "Prefrontal cortex-mediated executive function as assessed by Stroop task performance associates with weight loss among overweight and obese adolescents and young adults." Behavioural Brain Research, 321 (2017): 240-248. https://doi.org/10.1016/j.bbr.2016.12.040.
Sian Fitzpatrick et al. "Systematic Review: Are Overweight and Obese Individuals Impaired on Behavioural Tasks of Executive Functioning?." Neuropsychology Review, 23 (2013): 138-156. https://doi.org/10.1007/s11065-013-9224-7.
Author Response
We greatly appreciate your professional and patient comments. The references you provided are highly relevant to our research theme. Through reviewing these references, we found that incorporating some of them would add depth and comprehensiveness to our study. Based on your suggestions, we have made the following revisions:
-
Inclusion of "Michael J. Wheeler et al. "Distinct effects of acute exercise and breaks in sitting on working memory and executive function in older adults: a three-arm, randomised cross-over trial to evaluate the effects of exercise with and without breaks in sitting on cognition." British Journal of Sports Medicine, 54 (2019): 776 - 781. https://doi.org/10.1136/bjsports-2018-100168."
- This high-quality article found that a morning bout of moderate-intensity exercise improves serum BDNF levels and working memory or executive function in older adults, depending on whether subsequent sitting is interrupted with intermittent light-intensity walking. This conclusion provides valuable reference for how PTE can enhance executive functions in elderly populations.
- Inclusion of "Quintero Gacharná et al. "Acute effect of three different exercise training modalities on executive function in overweight inactive men: the brainfit study." Medicine and Science in Sports and Exercise, 50 (2018): 202. https://doi.org/10.1249/01.MSS.0000535750.40951.C1."
-
- This study compared the effects of acute resistance training(RT), high-intensity interval training (HIIT), and a combination of RT and HIIT. It found that even acute exercise can improve executive function in overweight individuals, with combined exercise showing better effects. This article, which we had previously overlooked, significantly enriches our study. Thank you for your suggestion, which has made our paper more comprehensive. We have included this study in our meta-analysis.
Additionally, we discovered that several articles initially included in our analysis were from the same project and needed to be consolidated. Therefore, we have revised the inclusion of literature by removing redundant studies and adding new ones. We have also reorganized the literature information, reprocessed the data, and updated the figures accordingly. Due to these changes, we have restructured our conclusion to make it more understandable and practical. The new version is likely to be more accessible and better suited for guiding practice: "This Network Meta-analysis found that Coordinated physical activity (CPA) and Aerobic exercise(AE) have significant benefits for cognitive flexibility, inhibitory control, and academic performance (AP) in children and adolescents. Besides, PTE improves executive functions (EFs) in adults and older adults. Combining the findings of this study with previous related research, we recommend that OW/OB begin by interrupting prolonged sedentary behavior and increasing fragmented physical activity, gradually incorporating AE, RT, and CPA (such as jump rope)."
Thank you again for your professional and patient advice, which has made our research more comprehensive